# Oxidative Stress and Antioxidant Defense in the Heart, Liver, and Kidney of Bat Species with Different Feeding Habits

**DOI:** 10.3390/ijms242216369

**Published:** 2023-11-15

**Authors:** Francielly Dias Pereira, Diego Antonio Mena Canata, Tiago Boeira Salomon, Fernanda Schäfer Hackenhaar, María João Ramos Pereira, Mara Silveira Benfato, Pabulo Henrique Rampelotto

**Affiliations:** 1Biophysics Department, Universidade Federal do Rio Grande do Sul, Porto Alegre 91501-970, Brazil; 2Post Graduate Program in Cellular and Molecular Biology, Universidade Federal do Rio Grande do Sul, Porto Alegre 91501-970, Brazil; 3Department of Medical Biosciences, Umeå University, 90185 Umeå, Sweden; 4Post Graduate Program in Animal Biology, Laboratory of Evolution, Systematics and Ecology of Birds and Mammals, Universidade Federal do Rio Grande do Sul, Porto Alegre 91501-970, Brazil; 5Graduate Program in Biological Sciences—Pharmacology and Therapeutics, Universidade Federal do Rio Grande do Sul, Porto Alegre 91501-970, Brazil; 6Bioinformatics and Biostatistics Core Facility, Instituto de Ciências Básicas da Saúde, Universidade Federal do Rio Grande do Sul, Porto Alegre 91501-970, Brazil

**Keywords:** diet, animal nutrition, oxidative stress, frugivorous, nectarivorous, insectivorous, hematophagous

## Abstract

The aim of this study was to compare the oxidative metabolism of four neotropical bat species with different feeding habits and investigate the relationship between their feeding habits and oxidative status. In terms of oxidative damage, our findings revealed major differences among the four bat species. In particular, hematophagous bats had lower levels of oxidative damage in the heart but higher levels in the liver. Nectarivorous bats had lower levels of carbonyl groups in the kidneys compared to insectivorous and hematophagous bats. The activity of various antioxidant and non-antioxidant enzymes in the heart, liver, and kidney also showed significant differences among the bat species. H_2_O_2_ consumption was lower in the heart of hematophagous bats, while insectivorous bats exhibited the highest enzymatic activity in the kidney. SOD activity was lower in the heart of hematophagous bats and lower in nectarivorous bats in the liver. Fumarase activity was higher in the heart of frugivorous/insectivorous and lower in nectarivorous/hematophagous bats. GPx activity was higher in the heart of nectarivorous/insectivorous and higher in the kidney of insectivorous bats. GST activity was higher in the heart of nectarivorous and lower in hematophagous bats. The correlation analysis between oxidative markers and enzymatic/non-enzymatic antioxidants in the heart, liver, and kidney exhibited distinct patterns of correlations due to variations in antioxidant defense mechanisms and oxidative stress responses in different organs. The observed differences in oxidative damage, antioxidant enzyme activities, and correlations between oxidative markers and antioxidants highlight the adaptability and complexity of the antioxidant defense systems in these bats. Each organ appears to have specific demands and adaptations to cope with oxidative stress based on its physiological functions and exposure to dietary components. Our results have major significance for the conservation and management of bats, which are threatened species despite being crucial components of ecosystems. Our study’s implications go beyond bat biology and offer valuable insights into comparative oxidative physiology.

## 1. Introduction

Bats are important ecological and evolutionary agents in the neotropical region. They are known to have diverse feeding habits and play a crucial role in reforestation (pollinators and seed dispersers) and pest control [1,2]. In recent years, studies have focused on the physiological and biochemical adaptations of bats to their feeding habits, including their oxidative metabolism. Antioxidants are molecules that protect cells from oxidative damage caused by reactive oxygen species and nitrogen species, which in excess can cause cell dysfunction and disease [3,4].

It is assumed that their oxidative metabolism is closely linked to their feeding habits, given the diverse range of diets they exhibit, which include nectarivorous, frugivorous, insectivorous, and hematophagous. Some bat species even combine two different diets, such as consuming both fruits and insects [5]. This diversity in diet among bats may be attributed to the unique physiological adaptations that allow them to exploit different food resources, highlighting their remarkable ecological versatility.

Frugivorous and nectarivorous bats are expected to have higher levels of antioxidants in comparison to their insectivorous and hematophagous counterparts [6]. This is because their diets are rich in carotenoids, flavonoids, and vitamins, which are known to be major antioxidants. On the other hand, the hematophagous diet contains high concentrations of iron due to its consumption of blood, making it highly oxidative [7]. The stark contrast in diets between these bat species highlights the importance of examining oxidative metabolism in relation to diet and ecological adaptations.

However, our knowledge of this relevant topic is limited due to the lack of comprehensive studies on the oxidative metabolism of a wide range of bat species, particularly those with different feeding habits. Most of the existing studies have focused on a small number of species or specific dietary groups, limiting our understanding of the overall variability in the antioxidant status of bats [8]. Furthermore, the physiology and metabolic pathways of bats are still poorly understood, and this can make it difficult to determine which antioxidants are most critical to the health and well-being of bats. Finally, most of the existing studies have focused on enzymatic antioxidants such as superoxide dismutase (SOD) and catalase (CAT) [9,10], and there is a lack of research on non-enzymatic antioxidants. Markers of oxidative damage in macromolecules together with non-enzymatic antioxidants may be more reliable indicators of oxidative metabolism than enzymatic antioxidants alone, knowing that non-enzymatic antioxidants have an important role in the neutralization of reactive species [4].

Considering these limitations, more comprehensive studies on oxidative metabolism in bats, particularly those with diverse feeding habits, are needed to provide a better understanding of the adaptations and mechanisms they use to cope with oxidative stress. Such studies may also have implications for the conservation and management of bats, which are crucial components of ecosystems and play an important role in maintaining ecosystem health and functioning.

In this work, we compared the oxidative metabolism of four neotropical male bat species with different feeding habits, namely: frugivorous, nectarivorous, insectivorous, and hematophagous. Our hypothesis is that the oxidative metabolism of these species will vary according to their feeding habits. To test our hypothesis, we measured the enzymatic antioxidant activities, non-enzymatic antioxidant levels, and oxidative damage in macromolecules (lipids and proteins) in three different organs (heart, liver, and kidney) of each species.

Studying the oxidative metabolism of various bat species can reveal their adaptive mechanisms and how they manage oxidative stress, which can shed light on the physiological adaptations that have enabled bats to thrive in diverse ecological niches. Moreover, our study’s implications go beyond bat biology and offer insights into comparative oxidative physiology.

## 2. Results

### 2.1. Oxidative Damage

Figure 1 illustrates the oxidative damage observed in the heart, liver, and kidney of four different bat-feeding habits. The exact *p*-value of each analysis is presented in Appendix A. The levels of carbonyl groups and malondialdehyde exhibited comparable patterns in the heart and liver (Figure 1A,B). Notably, hematophagous bats displayed significantly low levels of oxidative damage in the heart (0.019 ± 0.005 nmol/mg protein for carbonyl groups and 35.19 ± 10.89 nmol/mg protein for malondialdehyde) and high levels in the liver (1.1 ± 0.2 nmol/mg protein for carbonyl groups and 1317.5 ± 259.2 nmol/mg protein for malondialdehyde) when compared to the other species. In the kidneys, although the levels of malondialdehyde were relatively similar among bat species (3.0 ± 1.0 nmol/mg protein), the carbonyl group levels were significantly lower in nectarivorous bats (0.009 ± 0.006 nmol/mg protein) and higher in insectivorous (0.08 ± 0.03 nmol/mg protein) and hematophagous bats (0.11 ± 0.04 nmol/mg protein) (Figure 1A,B).

### 2.2. Antioxidant Enzymes

Figure 2 presents the activity of antioxidant enzymes measured in the heart, liver, and kidney of the four bat species. The exact *p*-value of each analysis is presented in Appendix A. The levels of H_2_O_2_ consumption, which measure the activity of enzymes, were found to be lower in the heart of hematophagous bats (995.5 ± 446.1 µmol/min/mg protein) compared to other bat species (Figure 2A left panel), while minimal variance was observed in the liver (13,269.6 ± 8645.4 µmol/min/mg protein) (Figure 2A middle panel). The kidney of insectivorous bats exhibited the highest enzymatic activity (14.0 ± 2.2 µmol/min/mg protein), while nectarivorous bats demonstrated comparatively lower activity levels (0.7 ± 0.4 µmol/min/mg protein) (Figure 2A right panel).

Regarding SOD, its activity was lower in the heart (2.7 ± 0.7 U/mg protein) and liver (2.6 ± 0.4 U/mg protein) of hematophagous bats (Figure 2B right panel; Figure 2B middle panel), and higher in frugivorous (2.8 ± 0.8 U/mg protein) and insectivorous bats (4.8 ± 0.9 U/mg protein) in the kidney (Figure 3B left panel).

The fumarase activity was higher in the heart of frugivorous (29.5 ± 7.6 U/mg protein) and insectivorous (26.0 ± 8.2 U/mg protein) while lower in nectarivorous (13.0 ± 9.3 U/mg protein) and hematophagous (6.6 ± 4.1 U/mg protein) (Figure 2C left panel); lower in nectarivorous (40.7 ± 27.7 U/mg protein) while higher in insectivorous (237.3 ± 42.6 U/mg protein) and hematophagous (203.0 ± 43.2 U/mg protein) in the liver (Figure 2C middle panel); and lower in nectarivorous (0.6 ± 0.3 U/mg protein) and frugivorous (0.5 ± 0.2 U/mg protein) and higher in insectivorous (5.4 ± 0.9 U/mg protein) and hematophagous (4.0 ± 0.9 U/mg protein) in the kidney (Figure 2C left panel).

The GPx activity was higher in the heart of nectarivorous (19,035.8 ± 4587.5 U/mg protein) and insectivorous (17,748.3 ± 3244.5 U/mg protein) (Figure 2D left panel); higher in frugivorous (20,027.7 ± 6185.6 U/mg protein) and insectivorous (21,368.6 ± 3096.0 U/mg protein) and lower in hematophagous (1573.2 ± 589.3 U/mg protein) in the liver (Figure 2D middle panel); and higher in the kidney of insectivorous bats (8.0 ± 1.8 U/mg protein) compared to other bat species (Figure 2D left panel).

Regarding GST activity, it was higher in the heart of nectarivorous (33.2 ± 8.4 U/mg protein) and lower in hematophagous (5.3 ± 1.6 U/mg protein) (Figure 2E left panel); higher in hematophagous (1127.0 ± 134.9 U/mg protein) and lower in nectarivorous (265.6 ± 81.1 U/mg protein) in the liver (Figure 2E middle panel); and higher in hematophagous (1.1 ± 0.3 U/mg protein) and lower in frugivorous (0.1 ± 0 U/mg protein) in the kidney (Figure 2E left panel).

### 2.3. Non-Enzymatic Antioxidants

Figure 3 presents the levels of non-enzymatic antioxidants measured in the heart, liver, and kidney of the four bat species. The exact *p*-value of each analysis is presented in Appendix A.

Total glutathione (Figure 3A), oxidized glutathione (Figure 3B), and reduced glutathione (Figure 3C) were lower in the heart of hematophagous bats compared to other bat species. In the heart, there was 508.6 ± 82.6 nmol/mg protein for total glutathione, 250.8 ± 69.2 nmol/mg protein for oxidized glutathione, and 348.3 ± 180.7 nmol/mg protein for reduced glutathione; in the liver, there was 3682.6 ± 1882.5 nmol/mg protein for total glutathione, 6776.2 ± 6007.7 nmol/mg protein for oxidized glutathione, and 418.3 ± 235.8 nmol/mg protein for reduced glutathione. In the kidney, they were lower in nectarivorous and hematophagous bats.

The GSSG/GSH ratio was found to be higher in the heart of hematophagous bats (0.85 ± 0.44) and lower in nectarivorous (1.15 ± 0.06) (Figure 3D left panel) and higher in the liver (59.28 ± 29.15) (Figure 3D middle panel) and kidney (0.53 ± 0.11) (Figure 3D right panel) of nectarivorous compared to other bat species.

The levels of nitrite and nitrates were lower in the heart (0.28 ± 0.11 nmol/mg protein) and kidney (2.9 ± 0.4 nmol/mg protein) of hematophagous compared to other bat species (Figure 3E left panel; Figure 3E right panel) and higher in the liver of frugivorous (6.0 ± 1.2 nmol/mg protein) and hematophagous (6.6 ± 2.5 nmol/mg protein) and lower in nectarivorous (2.2 ± 1.1 nmol/mg protein) and insectivorous (3.1 ± 0.4 nmol/mg protein) (Figure 3E middle panel).

The levels of Vitamin C were presented in a previous article recently published by our group [11].

### 2.4. Principal Component Analysis

Figure 4 presents the PCA made with the parameters measured in Figure 1, Figure 2 and Figure 3. Based on their feeding habitat, the PCA analysis results showed a distinct differentiation in the clustering of samples for the heart (Figure 4A), liver (Figure 4B), and kidney (Figure 4C). These results were confirmed by the pairwise PERMANOVA test among samples grouped according to bat species (Appendix A).

### 2.5. Correlation

The correlation between oxidative markers and (enzymatic and non-enzymatic) antioxidants measured in the heart, liver, and kidney of bats is presented in Figure 5. In the heart (Figure 5A), oxidative markers were positively associated with enzymatic and non-enzymatic antioxidants (except for GSSG/GSH). VitC was positively associated with enzyme activity, except for fumarase. In addition, most enzymes were positively correlated with each other. On the other hand, GSSG/GSH was negatively associated with oxidative markers, NO_2_ and NO_3_, and enzyme activity (except for fumarase and GPx).

In the liver (Figure 5B), the correlation pattern was more complex, with a mix of positive and negative associations. While carbonyl was positively associated with NO_2_ and NO_3_, MDA, and H_2_O_2_ consumption, it was negatively associated with VitC, SOD, and GPx. MDA was positively associated with NO_2_ and NO_3_, fumarase, and GST, but it was negatively associated with GSSG/GSH and SOD. SOD was negatively associated with oxidative markers, fumarase, and GST, but it was positively associated with GPx.

In the kidney (Figure 5C), the correlation pattern was more like the heart pattern, with carbonyl positively correlated with enzymatic activity and enzymes positively correlated with each other. Interestingly, GSSG/GSH was negatively associated with enzymatic activity (except for GST), VitC, and carbonyl.

## 3. Discussion

In this study, we measured and compared the oxidative metabolism of four neotropical bat species with different feeding habits and investigated the relationship between their feeding habits and oxidative status. Our results provide valuable insights into the oxidative damage and antioxidant defense mechanisms in these bat species.

Notably, while studies on wildlife usually face challenges in controlling confounding factors, we took careful measures to control major common interferences encountered in wildlife studies. All the animal species included in our research exhibit nocturnal habits and follow similar circadian cycles. They share similar foraging times, including feeding and flight activities. Additionally, we observed that all species had nearly identical entry and exit times for the caves. To ensure uniformity, the capturing process was carried out in the evening, ensuring that all bats had not consumed any food and were in the same basal physiological state. Furthermore, none of the four species engaged in hibernation or torpor, which are physiological adaptations that enable animals to conserve energy during unfavorable conditions. For these reasons, our observations indicate that the differences in these habits among the species are not significant enough to account for the variations we analyzed in terms of redox metabolism. Instead, we attribute these differences to variations in the diet of these species. Additionally, the stress associated with capturing the animals, including the time spent on the net, handling of the individuals, and euthanasia procedures, was consistent among all four species and was performed by the same individuals. By doing so, we ensured uniformity in the stressors applied, reducing the potential for significant intraspecies differences that could have arisen from these procedures.

Regarding the bat species, *Glossophaga soricina* is found throughout Latin America and consumes nectar flowers and floral parts [12]. This species is capable of sustained flight for extended periods. *Sturnira lilium* is found in South America and primarily feeds on fruits from the Solanaceae family [13]. They are adapted for flight and foraging in forested habitats. *Molossus molossus* is found in Latin America and feeds on various insects but shows a particular preference for Coleoptera [14]. This species is known for its fast and agile flight, allowing them to catch prey in flight. The common vampire bat *Desmodus rotundus* is also found in Latin America and is the only species that feeds on the blood of domestic cattle, which is high in protein but low in carbohydrates [15]. This requires them to have a larger body size and higher body weight compared to other bats (Table 1), which primarily feed on insects and fruits. Additionally, their larger size allows them to store more blood for longer periods between feedings. The four bat species are active at night and are considered nocturnal species.

In terms of oxidative damage, our findings revealed variations among the different bat species and organs. The levels of carbonyl groups and malondialdehyde, which serve as markers of oxidative damage, exhibited distinct patterns in the heart, liver, and kidney. Interestingly, hematophagous bats displayed low levels of oxidative damage in the heart but high levels in the liver. This finding suggests that hematophagous bats possess efficient antioxidant defense mechanisms in the heart, which may be attributed to their unique feeding habits and associated physiological adaptations. In contrast, the liver, being involved in the detoxification process, may experience increased oxidative stress due to the ingestion of blood meals rich in heme iron and other pro-oxidants [16].

Furthermore, the kidney exhibited differential patterns of oxidative damage across bat species. While malondialdehyde levels were relatively similar among the bat species, carbonyl group levels were lower in nectarivorous bats and higher in insectivorous and hematophagous bats. These differences could be attributed to variations in the metabolic demands of the kidney and the specific dietary components consumed by the different bat species [17]. Nectarivorous bats primarily consume plant-based nectar, which is rich in antioxidants, potentially contributing to their lower levels of oxidative damage in the kidney.

In addition to oxidative damage, we examined the activity of various antioxidant enzymes in the heart, liver, and kidney of the studied bat species. Our results demonstrated variations in the activity of antioxidant enzymes among the different feeding groups and organs. Hematophagous bats displayed lower activity of H_2_O_2_ consumption and SOD in the heart compared to other bat species, indicating potential adaptations to minimize oxidative stress in this vital organ [18]. Conversely, the kidney of insectivorous bats exhibited the highest enzymatic activity, suggesting an increased demand for antioxidant defense in this organ [19], possibly due to the higher metabolic rates associated with insectivory.

Non-enzymatic antioxidants also showed variations across the bat species and organs. The lower total glutathione levels in the heart of hematophagous bats suggest a reduced GSH pool in these bats. This may be indicative of lower baseline antioxidant capacity in their hearts, potentially making them more susceptible to oxidative stress [20]. The liver of all bat species showed higher GSH levels compared to the heart. This is in line with the liver’s role as a major organ for GSH synthesis and storage. The higher GSH levels in the liver contribute to maintaining the overall GSH pool in bats, which is essential for antioxidant defense and detoxification processes [21]. The lower GSH levels in the kidney of nectarivorous and hematophagous bats indicate a potential reduction in the GSH pool in these bat species. This may have implications for their ability to counteract oxidative stress and detoxify harmful compounds in the kidney. Hematophagous bats displayed higher GSSG/GSH ratios in the heart, indicating an imbalance in the redox state, potentially due to the presence of pro-oxidants from blood meals [22]. Interestingly, nectarivorous bats exhibited higher GSSG/GSH ratios in the liver and kidney, suggesting a higher demand for GSH recycling and antioxidant capacity in these organs. The GSSG/GSH ratio is an important indicator of oxidative stress and redox balance, and its elevation in specific organs reflects the dynamic nature of antioxidant defense mechanisms in bats [23].

Our PCA results further supported the differentiation of samples based on their feeding habits in the heart, liver, and kidney. This finding suggests that feeding habits play a significant role in shaping the antioxidant profiles and oxidative status of bats, highlighting the influence of dietary components on oxidative metabolism.

The correlation analysis between oxidative markers and enzymatic/non-enzymatic antioxidants provided additional insights into the relationships between oxidative stress and antioxidant defense in the bat species studied. These correlations can shed light on the interplay between antioxidant capacity and oxidative damage, further elucidating the adaptive strategies employed by bats to cope with oxidative stress induced by their respective feeding habits [24]. Overall, the heart, liver, and kidney exhibited different patterns of correlations due to variations in antioxidant defense mechanisms and oxidative stress responses in different organs. The mixed positive and negative associations highlight the complex nature of antioxidant defense mechanisms and oxidative stress responses in these tissues. It further emphasizes the importance of studying organ-specific variations in oxidative stress and antioxidant systems to gain a comprehensive understanding of the underlying mechanisms.

In the heart, the positive associations highlight potential adaptive responses, where increased oxidative marker levels trigger enhanced antioxidant defense mechanisms. The positive associations between VitC and enzyme activity further emphasize their complementary roles in mitigating oxidative stress [25]. The negative association between GSSG/GSH and oxidative markers could indicate that when the oxidative stress is high, the GSH in the cells is being consumed, leading to a decrease in GSH levels and an increase in GSSG/GSH ratio. The negative association between GSSG/GSH and enzyme activity could also be explained by the fact that some enzymes require GSH as a cofactor for their activity. For example, GPx is an enzyme that uses GSH to detoxify H_2_O_2_ and lipid peroxides, and a decrease in GSH levels could lead to a decrease in GPx activity, which in turn would increase oxidative stress. Therefore, the negative association between GSSG/GSH and the other measured parameters could indicate a shift in the redox balance towards oxidation, which may have negative consequences for cellular function and health [26].

In the liver, the positive correlations of carbonyl are related to elevated levels of nitrogen species, MDA, and increased H_2_O_2_ consumption, reflecting oxidative stress [27]. On the other hand, the negative correlations of carbonyl indicate a potential depletion of antioxidant defenses in response to increased oxidative stress [28]. Regarding MDA, its positive correlations imply that higher MDA levels are linked to increased levels of nitrogen species, fumarase, and GST activity, suggesting oxidative stress and potential adaptive responses [29]. On the other hand, its negative associations suggest impaired antioxidant defenses and potential oxidative damage due to elevated MDA levels. SOD activity was associated with lower levels of oxidative markers, fumarase, and GST, suggesting an antioxidant role of SOD in mitigating oxidative stress [30]. On the other hand, SOD activity was linked to increased levels of GPx, highlighting potential cooperative effects between these enzymes in antioxidant defense mechanisms [31].

In the kidney, the correlation pattern was more similar to the heart. Carbonyl in the kidney was positively correlated with enzymatic activity [32], similar to the heart pattern. Enzymes in the kidney also show positive correlations with each other, indicating a coordinated response of enzymatic antioxidants [33]. Interestingly, GSSG/GSH in the kidney was negatively associated with enzymatic activity (except for GST), VitC, and carbonyl. This suggests that higher GSSG/GSH levels in the kidney are associated with decreased enzymatic antioxidant activity, lower VitC levels, and potentially increased oxidative stress represented by carbonyl levels.

These findings contribute to our understanding of the intricate relationship between feeding habits, oxidative metabolism, and antioxidant defense in neotropical bat species. Furthermore, they underscore the importance of considering multiple biomarkers and organs when assessing the antioxidant status of bats, as different organs may respond differently to oxidative stress depending on their functional roles and exposure to dietary pro-oxidants. Bats’ varying restrictive diets not only shape their morphological structures but also modulate their metabolic patterns [34]. These adaptations enable them to maintain normal energetic processes during flight while minimizing adverse oxidative effects. It is noteworthy that despite experiencing high metabolic peaks, bats seem to have minimal detrimental effects due to oxidation. This phenomenon suggests a unique metabolic regulation system in bats that effectively mitigates the adverse effects of oxidation. The ability to maintain a high level of metabolic activity with minimal oxidative stress highlights the impressive physiological adaptations of bats and underscores the significance of studying their metabolic processes.

In addition, the life expectancy of bats can vary due to their diet. Different species of bats have different dietary preferences, and their diet can impact their overall health and lifespan [35]. Bats that consume a variety of nutritionally rich diets tend to have longer lifespans than those with limited or poor-quality food sources. For example, insectivorous bats tend to have relatively shorter lifespans compared to other bat species. Their high metabolic rate, which is necessary for active flight and capturing fast-moving insects, can lead to increased wear and tear on their bodies over time. As a result, many insectivorous bats have relatively shorter lifespans, typically ranging from 2 to 10 years, although some may live longer. On the other hand, frugivorous and nectarivorous bats generally have longer lifespans compared to insectivorous bats [36]. The diet of fruit bats is often composed of fruits and nectar, which are energy-rich and less demanding on their metabolism compared to chasing insects. This can lead to longer lifespans, with some fruit bats living up to 20–30 years in the wild. In turn, hematophagous bats may have a life expectancy of around 7 to 12 years in the wild. As blood is a high-quality food source, hematophagous bats require less energy to obtain their nutrients than insectivorous bats, for example, which need to consume large quantities of insects to meet their dietary needs [37]. Additionally, hematophagous bats have evolved specialized adaptations, such as powerful jaw muscles and sharp teeth, which allow them to feed on blood more efficiently [38]. These adaptations help reduce the amount of energy they need to expend on hunting and feeding, which in turn may contribute to their longer lifespan. Additionally, other factors, such as predation, habitat availability, and environmental conditions, can also affect the life expectancy of bats. Bats face various challenges in the wild, and their longevity is influenced by a combination of factors, including diet, ecological niche, and environmental conditions.

Our study’s limitation is the relatively small sample size of animals per group, which could presume a limited generalization of our findings. However, obtaining a larger sample size was challenging due to the difficulties in capturing and studying wildlife animals, especially for threatened species such as bats. We were only authorized to use 10 animals per group under the planned experimental design, as per Brazilian law’s strict regulations on capturing wildlife for research purposes. Another important aspect to be discussed is seasonality. Similar to other studies comparing different bat diets [16], our work encompassed various seasons of the year, with each bat species collected in a specific season. Factors influencing food availability, including seasonality and weather, may affect the diet of each bat species differently. Seasonal activity patterns of bats have been observed in various studies [39,40,41,42], but it is important to note that not all bat species show seasonal variation in their diets. For instance, a previous study on *S. lilium*, the frugivorous bat analyzed in our work, did not observe seasonal variation in its diet [13]. These findings highlight the highly variable and complex nature of seasonal activity patterns in bats. In our study, insectivorous and hematophagous bats were collected during the same summer season, yet they presented different PCA profiles. This suggests that while seasonality is a factor, it probably does not explain the heterogeneity found in our results. Diet likely plays a crucial role in shaping the observed differences.

Future research endeavors should aim to investigate the underlying molecular mechanisms responsible for the observed variations in antioxidant status among bat species with different feeding habits. Additionally, longitudinal studies examining the antioxidant profiles and oxidative stress responses of bats in different environmental contexts and physiological conditions will enhance our understanding of the dynamic nature of antioxidant defense systems. In addition, studies including both male and female bats should be performed to investigate potential gender-specific differences in their metabolic traits.

Overall, this study contributes to the growing body of knowledge on the antioxidant status of bats and emphasizes the need for further research to fully comprehend the adaptive strategies and physiological trade-offs involved in maintaining redox balance in these remarkable neotropical bat species. Such insights are crucial for conservation efforts, as understanding the antioxidant capacity of bats can inform strategies to mitigate oxidative stressors in their natural habitats and promote their overall health and well-being.

## 4. Materials and Methods

### 4.1. Ethical Aspects

The bats in this study were captured under a license authorized by the Brazilian Biodiversity Information and Authorization System (SISBIO, No. 47202-1) and the National Council for the Control of Animal Experimentation (CONCEA, No. 33339). In addition, the Ethics Committee on the Use of Animals of the Federal University of Rio Grande do Sul approved the study (No. 28645).

### 4.2. Animals and Organ Collection

The four bat species used in this study and their feeding habits are presented in Table 1. The bat species captured were *Glossophaga soricina* (*n* = 10), *Sturnira lilium* (*n* = 10), *Molossus molossus* (*n* = 10), and *Desmodus rotundus* (*n* = 9). Appendix A indicates the collection sites and coordinates. Details of the capturing process of all animals as well as organ removal and storage were described in our previous study [43]. Briefly, 39 adult male bats were captured in southern Brazil from summer 2018 to winter 2019. In the field or with voucher specimens, estimating the degree of ossification in wing elements is an established technique for distinguishing adult bats [44], which was the technique used in our study. Different capturing methods, such as dip nets, mist nets, or harp traps, were used based on the specific shelter type. Bats were captured at the beginning of the night to ensure fasting and were on-site euthanized using an intraperitoneal injection of xylazine (10 mg/kg) and ketamine (60 mg/kg). For this reason, it was not possible to collect blood samples. After euthanasia, the animals were immediately placed in plastic bags, frozen in liquid nitrogen, and kept on dry ice until their subsequent transfer to the local facility at the Biophysics Department, Universidade Federal do Rio Grande do Sul (Porto Alegre 91501-970, Rio Grande do Sul, Brazil), where they were stored in a freezer at −80 °C.

### 4.3. Organ Processing

The heart, liver, and kidneys were manually macerated. The tissues were macerated in a solution of 10 mL K_3_PO_4_ buffer (30 mmol/L), potassium chloride (120 mmol/L), phenylmethylsulfonyl fluoride (0.201 mmol/L), and desferroxamine (1.5 mmol/L). The samples were then sonicated three times for 10 s, followed by centrifugation at 1700× *g* twice for 10 min. The final supernatant was aliquoted into 1.5 mL microtubes and stored in a freezer at −80 °C. A 14,000× *g* centrifugation was carried out for 5 min before each assay.

### 4.4. Biochemical Analysis

Details of the biochemical analysis are presented in Appendix A. Briefly, oxidative damage was assessed by the levels of carbonyl groups and malondialdehyde (MDA). In addition, the activity of the following enzymes was measured: superoxide dismutase (SOD), fumarase, glutathione peroxidase (GPx), glutathione-S-transferase (GST), and H_2_O_2_ consumption. Non-enzymatic antioxidants were assessed by the amounts of nitrate and nitrite (NO2 and NO3) and the GSSG/GSH ratio.

## 5. Conclusions

In conclusion, our study provides comprehensive insights into the oxidative metabolism of neotropical bat species with different feeding habits. The observed variations in oxidative damage, enzymatic and non-enzymatic antioxidant activities, and correlations between oxidative markers and antioxidants highlight the complexity and adaptability of the antioxidant defense systems in these bats. The distinct patterns observed in oxidative damage and antioxidant enzyme activities among the different organs and bat species suggest that each organ has its own specific demands and adaptations to cope with oxidative stress based on its physiological functions and exposure to different dietary components.

## Figures and Tables

**Figure 1 ijms-24-16369-f001:**
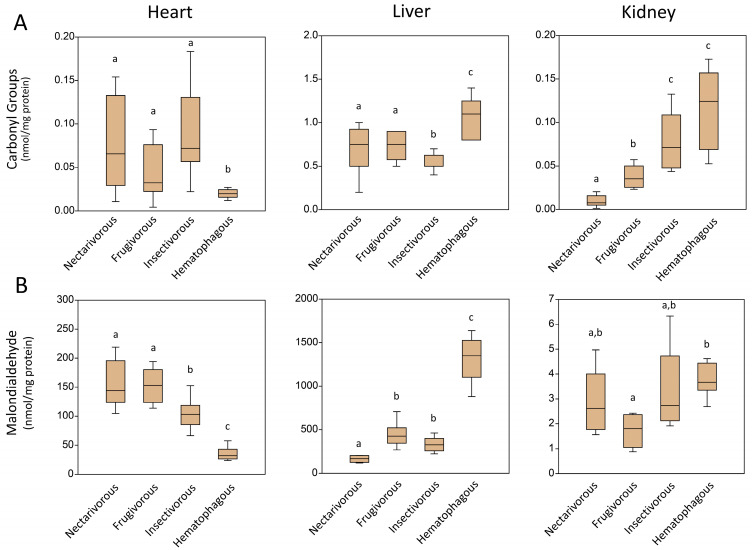
Levels of oxidative damage markers measured in the heart, liver, and kidney of nectarivorous, frugivorous, insectivorous, and hematophagous bats. Data are presented as the median (interquartile range). Different letters indicate statistical differences among species (*p* < 0.05). The same letters correspond to no statistical differences (*p* > 0.05). The exact *p*-value of each analysis is presented in Appendix A.

**Figure 2 ijms-24-16369-f002:**
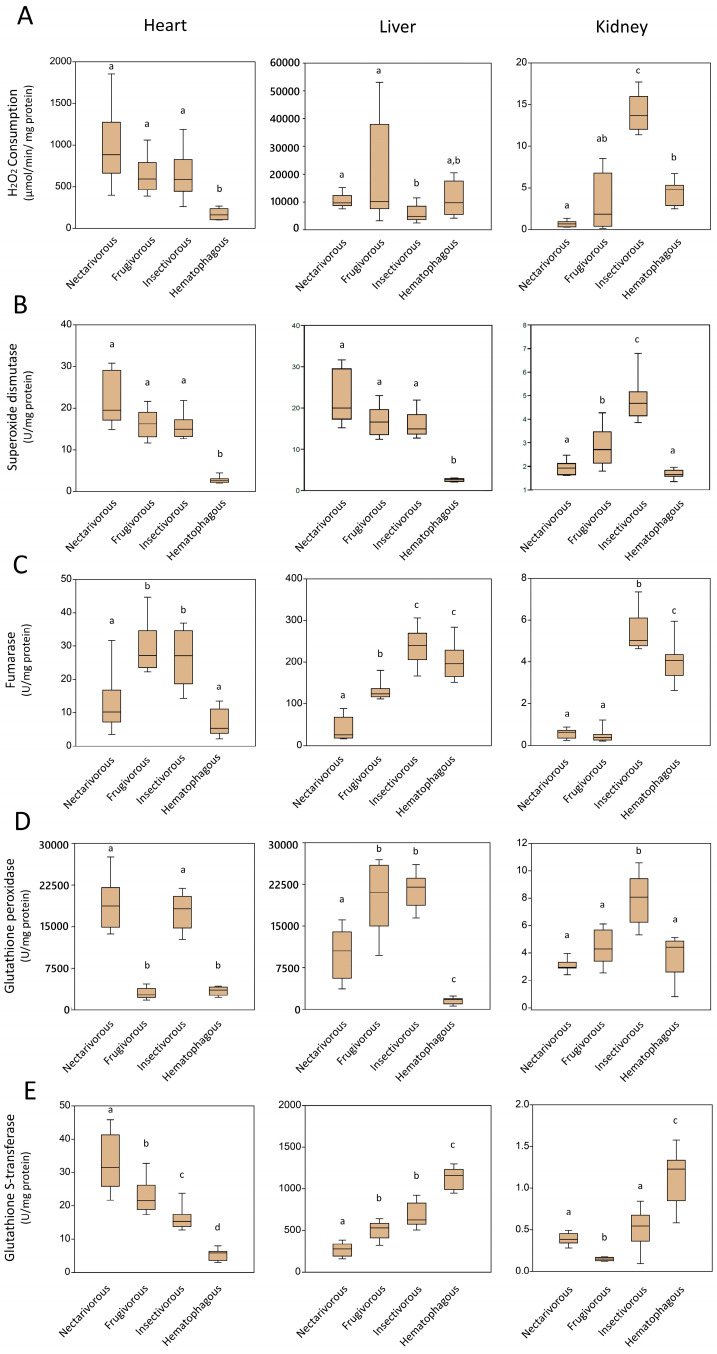
The activity of antioxidant enzymes measured in the heart, liver, and kidney of nectarivorous, frugivorous, insectivorous, and hematophagous bats. (**A**) H_2_O_2_ consumption; (**B**) Superoxide dismutase; (**C**) Fumarase; (**D**) Glutathione peroxidase (**E**) Glutathione S-transferase. Data are presented as the median (interquartile range). Different letters indicate statistical differences among species (*p* < 0.05). The same letters correspond to no statistical differences (*p* > 0.05). The exact *p*-value of each analysis is presented in Appendix A.

**Figure 3 ijms-24-16369-f003:**
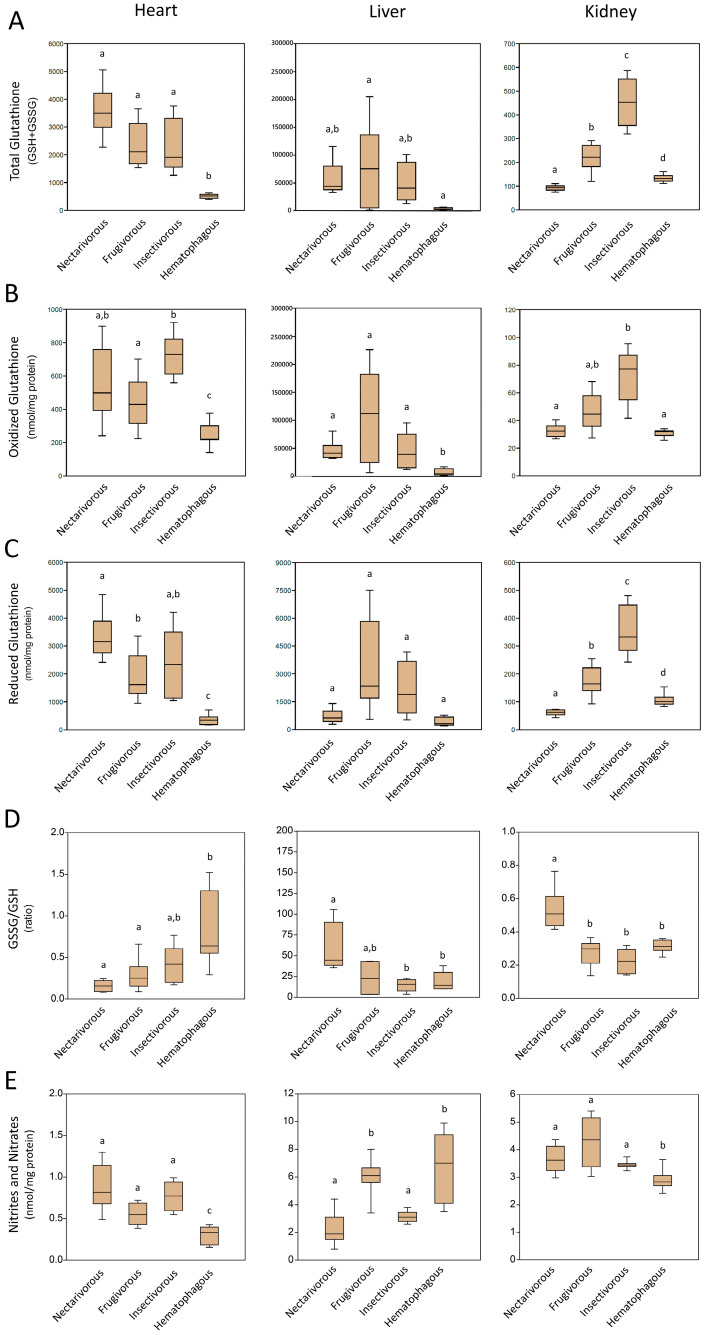
Levels of non-enzymatic antioxidants measured in the heart, liver, and kidney of nectarivorous, frugivorous, insectivorous, and hematophagous bats. (**A**) Total Glutathione; (**B**) Oxidized Glutathione; (**C**) Reduced Glutathione; (**D**) GSSG/GSH; (**E**) Nitrites and Nitrates. Data are presented as the median (interquartile range). Different letters indicate statistical differences among species (*p* < 0.05). The same letters correspond to no statistical differences (*p* > 0.05). The exact *p*-value of each analysis is presented in Appendix A.

**Figure 4 ijms-24-16369-f004:**
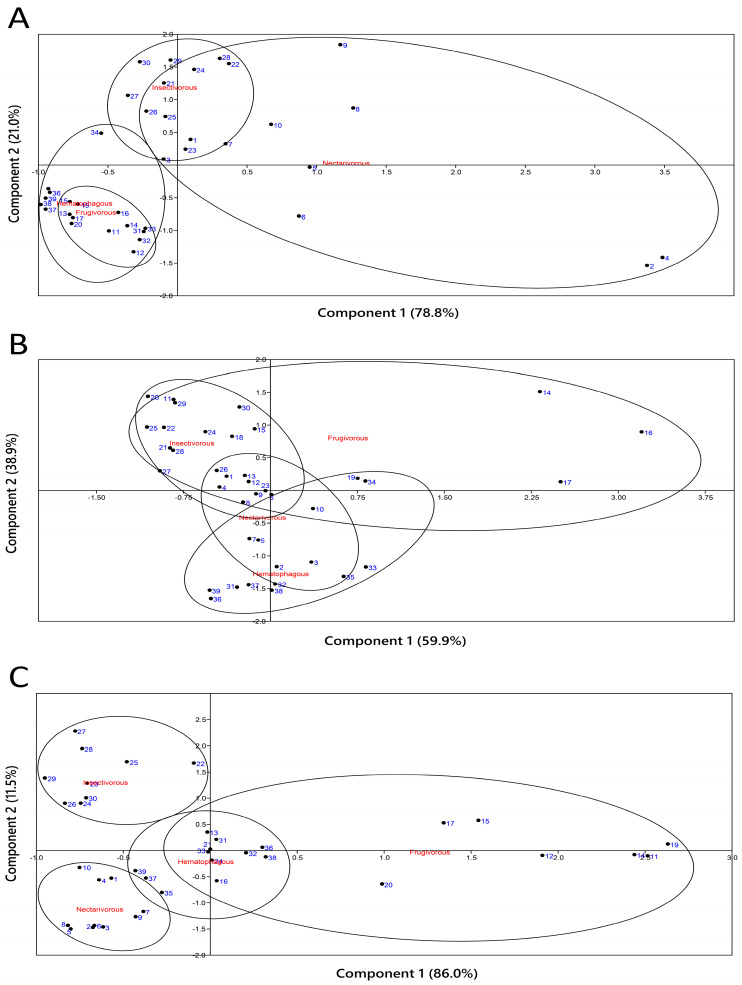
Principal component analysis of oxidative markers and (enzymatic and non-enzymatic) antioxidants measured in the heart, liver, and kidney of nectarivorous, frugivorous, insectivorous, and hematophagous bats. Numbers in blue indicate samples. (**A**) Heart; (**B**) liver; (**C**) kidney.

**Figure 5 ijms-24-16369-f005:**
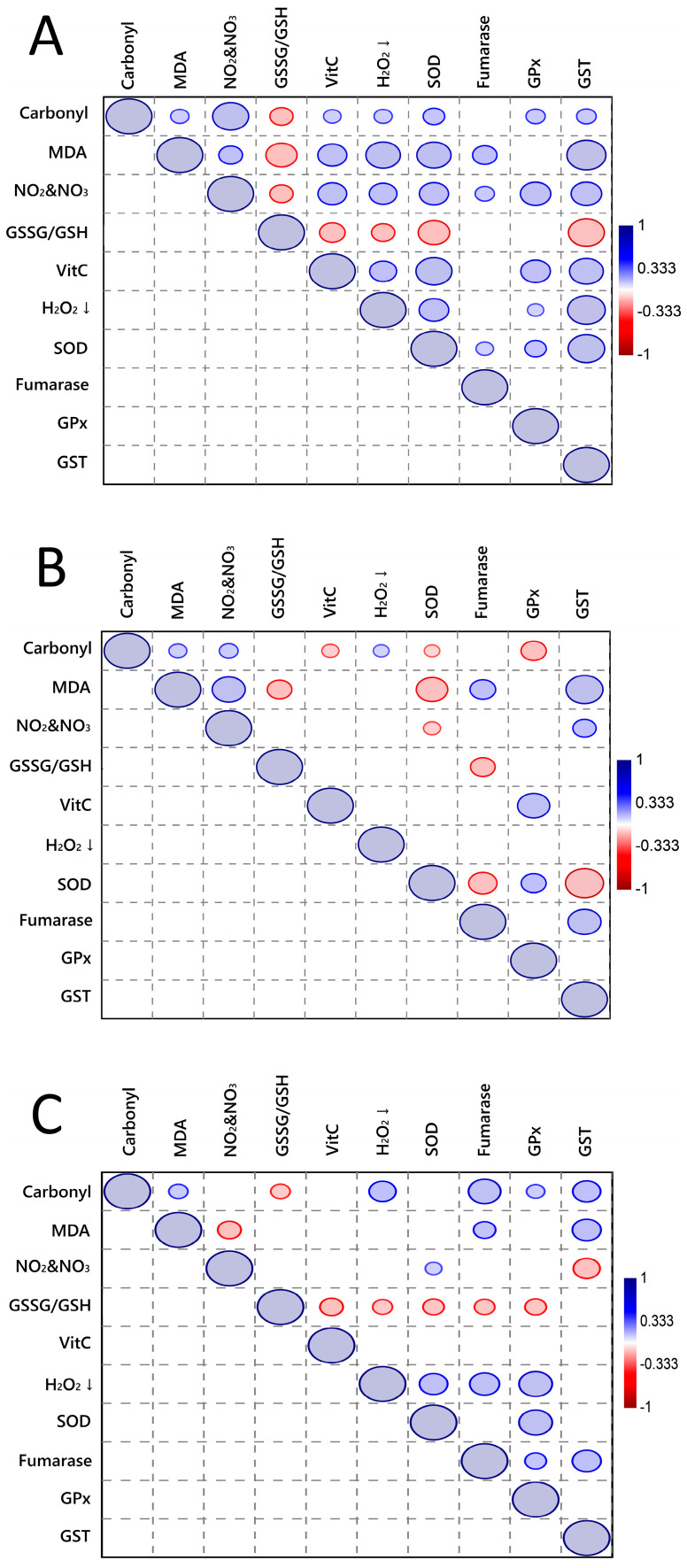
Graphical Spearman’s correlation matrix of oxidative markers and (enzymatic and non-enzymatic) antioxidants measured in this study. (**A**) Heart; (**B**) liver; (**C**) kidney. Positive correlation (from white to blue); negative correlation (from white to red). Only significant correlations are presented (*p* < 0.05). The exact *p*-value of each correlation is presented in Appendix A (heart), Appendix A (liver), and Appendix A (kidney). The circle size represents the correlation coefficient. H_2_O_2_↓ means hydrogen peroxide consumption.

**Table 1 ijms-24-16369-t001:** Biological data of each species, including feeding habits, body weight, and weight of each organ.

Bat Species	Feeding Habit	Season	Body Weight (g)	Heart (g)	Liver (g)	Kidneys (g)
*Glossophaga soricina*	Nectarivorous	Autumn (2019)	16.81 ± 1.33	0.46 ± 0.07	0.34 ± 0.06	0.13 ± 0.01
*Sturnira lilium*	Frugivorous	Winter (2019)	21.60 ± 1.57	0.41 ± 0.08	0.81 ± 0.11	0.29 ± 0.06
*Molossus molossus*	Insectivorous	Summer (2018)	18.09 ± 1.80	0.41 ± 0.04	0.55 ± 0.05	0.14 ± 0.01
*Desmodus rotundus*	Hematophagous	Summer (2018)	40.85 ± 3.15	0.79 ± 0.18	1.84 ± 0.23	0.48 ± 0.04

## Data Availability

Data are contained within the article and Appendix A.

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
