# Peer review of "Oxidative Stress and Antioxidant Defense in the Heart, Liver, and Kidney of Bat Species with Different Feeding Habits"

_ijms, 2023, doi:10.3390/ijms242216369_

Round 1

Reviewer 1 Report (New Reviewer)

Comments and Suggestions for Authors

Introduction to the topic of work

Oxidative stress is a state of imbalance between the formation of free radicals and the body’s antioxidant abilities. It causes numerous damage to the body, including lipid peroxidation, DNA and protein damage. In order to counteract the effects of oxidative stress, the body has developed defense mechanisms based on the capture of free radicals, inhibition of their formation and chelation of transition metal ions, which catalyze free radical reactions. Free radicals of an oxygen or nitrogen atom with unpaired electrons are capable of participating in chemical reactions. The presence of an unpaired electron makes these molecules highly reactive, as they tend to pair electrons by receiving or giving them away. The free radical mainly includes oxygen compounds ROS (reactive oxygen species) and nitrogen RNS (reactive species). Among the free radicals can be distinguished: nitrogen, superoxide anionic radical (O2·– ), hydroxyl radical (OH-), ROO peroxides and LOO lipid peroxides. Some molecules, i. e. : hydrogen peroxide (H2O2), ozone (O3), singlet oxygen (O2 ), chloric acid (I) (HOCl), peroxynitrate (V) (ONOO- ) are not free radicals, but are classified as oxidants because they very easily cause free radical reactions in living organisms. Against too high a concentration of free radicals against the toxic effects of ROS, protects the antioxidant system. The main task of the body’s defense mechanisms is to neutralize free radicals, inhibit free radical chain reactions and protect the cell from their toxic effects.

The aim of the study was to evaluate the oxidative metabolism of four neotropic male bats with different feeding habits, namely: fruit-eating, nectar-eating, insect-eating and haematophageal, and the effect on antioxidant enzymatic activity, non-enzymatic antioxidant levels and oxidative damage in macro-molecules (lipid and protein) in three different organs (heart, liver and kidney) of each species.

Specific comments

Abstract

Complete the abstract after giving the most important results in numerical form.

Methodology

Lack of relevant information in the methodology, e. g. age of bats that were euthanized and further analysis?

No description/analysis of diet of bats in the study?

What time of year were the tests conducted?

Do I need to complete a manuscript on how to harvest and store organs?

Why wasn't the blood tested?

Results

Results should be given in the text in numerical form and expressed as statistically significant

Discussion

The discussion should include literature references on foraging time, diet, flying time and age.

Summary

The article needs to be thoroughly improved and supplemented.

Author Response

Introduction to the topic of work

Oxidative stress is a state of imbalance between the formation of free radicals and the body’s antioxidant abilities. It causes numerous damage to the body, including lipid peroxidation, DNA and protein damage. In order to counteract the effects of oxidative stress, the body has developed defense mechanisms based on the capture of free radicals, inhibition of their formation and chelation of transition metal ions, which catalyze free radical reactions. Free radicals of an oxygen or nitrogen atom with unpaired electrons are capable of participating in chemical reactions. The presence of an unpaired electron makes these molecules highly reactive, as they tend to pair electrons by receiving or giving them away. The free radical mainly includes oxygen compounds ROS (reactive oxygen species) and nitrogen RNS (reactive species). Among the free radicals can be distinguished: nitrogen, superoxide anionic radical (O2·– ), hydroxyl radical (OH-), ROO peroxides and LOO lipid peroxides. Some molecules, i. e. : hydrogen peroxide (H2O2), ozone (O3), singlet oxygen (O2 ), chloric acid (I) (HOCl), peroxynitrate (V) (ONOO- ) are not free radicals, but are classified as oxidants because they very easily cause free radical reactions in living organisms. Against too high a concentration of free radicals against the toxic effects of ROS, protects the antioxidant system. The main task of the body’s defense mechanisms is to neutralize free radicals, inhibit free radical chain reactions and protect the cell from their toxic effects.

The aim of the study was to evaluate the oxidative metabolism of four neotropic male bats with different feeding habits, namely: fruit-eating, nectar-eating, insect-eating and haematophageal, and the effect on antioxidant enzymatic activity, non-enzymatic antioxidant levels and oxidative damage in macro-molecules (lipid and protein) in three different organs (heart, liver and kidney) of each species.

Reply: We thank the reviewer for the useful comments that helped to improve the current version of the manuscript. We have carefully analyzed each comment and all related changes in the text of the manuscript were highlighted in yellow to facilitate revision. We hope we have properly addressed all suggestions raised by the reviewer. Any additional information can be immediately provided.

Specific comments

Abstract

Complete the abstract after giving the most important results in numerical form.

Reply: We have completed the abstract giving the most important results (similar to our previous article (https://www.mdpi.com/1422-0067/24/15/12162), but numerical form was not included due to word limitation and to avoid unnecessary complexity. However, we have included all numerical information in the text and supplementary tables.

Methodology

Lack of relevant information in the methodology, e. g. age of bats that were euthanized and further analysis?

Reply: We have included this information in the manuscript with proper reference (lines 435-446).

No description/analysis of diet of bats in the study?

Reply: This information was already described in the third paragraph of the Discussion section. Please, let us know if the reviewer finds this information better suited to the Methods Section.

What time of year were the tests conducted?

Reply: We have included this information in the manuscript and Table 1. Bats were from summer 2018 to winter 2019.

Do I need to complete a manuscript on how to harvest and store organs?

Reply: Sorry for this forgetfulness. We have included this information in the manuscript. (lines-435-446).

Why wasn't the blood tested?

Reply: It was not possible to collect blood because bats were euthanized on-site and were immediately frozen in liquid nitrogen to preserve the organs. This information was included in the manuscript. 440-443).

Results

Results should be given in the text in numerical form and expressed as statistically significant.

Reply: We have included this information in the manuscript.

Discussion

The discussion should include literature references on foraging time, diet, flying time and age.

Reply: We complemented this information in the manuscript, which already is provided in the listed references. In addition, several new references were included in the discussion.

Summary

The article needs to be thoroughly improved and supplemented.

Reply: We hope we have properly addressed all suggestions raised by the reviewer. Any additional information can be immediately provided.

Reviewer 2 Report (New Reviewer)

Comments and Suggestions for Authors

Dear Editor, Dear Authors,

I enjoyed reading this document. Ecology is not my field, but the biochemical approaches are rigorous and make sense. The general message is interesting and will be of interest for the reader of the field and I’m convinced for others studying oxidative metabolism in more conventional models. I’ve some comments about structure of the manuscript and about discussion which could improve a little the comprehension and the discussion.

First, I’m convinced that the mat & meth must be completed by the capture protocol and by the statistical part. This is two important part which must be present in the main text. Indeed, the fact that bats were captured at the evening in fasting period is an important point for analysis of metabolic organ. This is only indicated at the end of the discussion.

PC component analyses are interesting, but I think must be displayed at the end of the results part, because they are only another presentation of the different results already obtained.

Presentation of the results is short but of sufficient quality. I think that the authors could precise two points. First the definition of the letter used for statistical difference must be indicated in all the figure legends, and it could be precise that same letters correspond to no statistical difference. Indeed, this representation is not sufficiently generalized to be directly understand by the reader. Moreover, the fact that p value was displayed for figure 5 in supplementary is perfect, but must be added for Figure 2, 3 and 4 to.

GSH/GSSG ratio is an important value. Nevertheless, the authors could display in addition the quantity of GSH and of GSH+GSSG as indicators of GSH pool and GSH neosynthesis, data already measured following the protocol used by the authors. This is an important point which could support and complete the discussion part about GSH.

The discussion is long but highly interesting. Nevertheless, several lacks exist:

- the authors must discuss more their results with others existing analyses, as their own for brain of the same bats or with other analysis about specific population of bats.

- the discussion is not sufficiently referenced. All assertion must be referenced.

- I mind that two important points have not been addressed. The first is the fact that between species, especially when the diet is different, the general metabolism is different and influences the oxidative metabolism of organs. For laboratory animals, it's easy to analyze the RER to determine whether they prefer to oxidize carbohydrates or lipids, or prefer to use "anaerobic" glycolysis. Is there a difference between these bat species on these points? Perhaps an analysis of the mitochondrial content of the various organs could shed light on this difference.

The second is about seasonality. Bats have been captured on a period of one year and a half, a time covering different seasons. Does the date of the capture have influenced the results in a same species and explained the heterogeneity found? This could seems especially true for frugivorous and nectarivorous bat species which can be dependent to the seasonality of plants. I’m aware about the low number of samples and the difficulty to obtain it, but I think that this point merit to be discussed and take into account to analyse the results.

Author Response

Dear Editor, Dear Authors,

I enjoyed reading this document. Ecology is not my field, but the biochemical approaches are rigorous and make sense. The general message is interesting and will be of interest for the reader of the field and I’m convinced for others studying oxidative metabolism in more conventional models. I’ve some comments about structure of the manuscript and about discussion which could improve a little the comprehension and the discussion.

Reply: Reply: We thank the reviewer for the useful comments that helped to improve the current version of the manuscript. We have carefully analyzed each comment and all related changes in the text of the manuscript were highlighted in yellow to facilitate revision. We hope we have properly addressed all suggestions raised by the reviewer. Any additional information can be immediately provided.

First, I’m convinced that the mat & meth must be completed by the capture protocol and by the statistical part. This is two important part which must be present in the main text. Indeed, the fact that bats were captured at the evening in fasting period is an important point for analysis of metabolic organ. This is only indicated at the end of the discussion.

Reply: We tried to include as much information as possible on these aspects (lines-435-446). But any additional information can be immediately provided.

PC component analyses are interesting, but I think must be displayed at the end of the results part, because they are only another presentation of the different results already obtained.

Reply: We agree and have included the PCA after the individual presentation of oxidative damage and antioxidant markers but before the correlations (like our previous manuscript focused on the brain). PCA now is in Figure 4.

Presentation of the results is short but of sufficient quality. I think that the authors could precise two points. First the definition of the letter used for statistical difference must be indicated in all the figure legends, and it could be precise that same letters correspond to no statistical difference. Indeed, this representation is not sufficiently generalized to be directly understand by the reader. Moreover, the fact that p value was displayed for figure 5 in supplementary is perfect, but must be added for Figure 2, 3 and 4 to.

Reply: We have indicated the definition of the letter used for statistical difference/no difference as well as the supplementary tables with p-values for Figure 2, 3 and 4.

GSH/GSSG ratio is an important value. Nevertheless, the authors could display in addition the quantity of GSH and of GSH+GSSG as indicators of GSH pool and GSH neosynthesis, data already measured following the protocol used by the authors. This is an important point which could support and complete the discussion part about GSH.

Reply: We have included the Total Glutathione (GSH+GSSG), Oxidized Glutathione, and Reduced Glutathione and have properly discussed the results as indicators of GSH pool and GSH neosynthesis. See Figure 3A, B, and C… and Discussion lines 287-296.

The discussion is long but highly interesting. Nevertheless, several lacks exist:

- the authors must discuss more their results with others existing analyses, as their own for brain of the same bats or with other analysis about specific population of bats.

Reply: We tried to improve this part as much as possible and included a series of new references.

- the discussion is not sufficiently referenced. All assertion must be referenced.

Reply: We have included several new references in the discussion.

- I mind that two important points have not been addressed. The first is the fact that between species, especially when the diet is different, the general metabolism is different and influences the oxidative metabolism of organs. For laboratory animals, it's easy to analyze the RER to determine whether they prefer to oxidize carbohydrates or lipids, or prefer to use "anaerobic" glycolysis. Is there a difference between these bat species on these points? Perhaps an analysis of the mitochondrial content of the various organs could shed light on this difference.

Reply: You raise an important point about the potential differences in general metabolism between bat species, especially in relation to their diet and how it may influence their oxidative metabolism. While the current study did not specifically investigate the RER or mitochondrial content of the organs, it is possible that such analyses could provide further insights into the differences observed in the activity of antioxidant enzymes between the bat species. Future studies could explore these aspects to better understand the metabolic adaptations of different bat species and how they relate to their physiology and ecology. Thank you again for your valuable input.

The second is about seasonality. Bats have been captured on a period of one year and a half, a time covering different seasons. Does the date of the capture have influenced the results in a same species and explained the heterogeneity found? This could seems especially true for frugivorous and nectarivorous bat species which can be dependent to the seasonality of plants. I’m aware about the low number of samples and the difficulty to obtain it, but I think that this point merit to be discussed and take into account to analyse the results.

Reply: That’s also an interesting point to be discussed. While we cannot answer this question for sure (because the same bat species was not collected in different seasons for comparison), our results indicate that seasonality has not explained the heterogeneity found because Insectivorous and Hematophagous were collected during the same summer season but presented different PCA profiles in the three comparisons (Fig. XA, A, and C). However, seasonality is observed in different bat species, so the species in our study may have seasonal variations in their diet and biochemical parameters (which could be the focus of a future study). We have included the season in which each species was collected in Table 1, and have included this discussion in lines 395-407.

Round 2

Reviewer 1 Report (New Reviewer)

Comments and Suggestions for Authors

The article has been substantially revised and supplemented in accordance with the comments to its previous version.

I would also ask you to detail the diet of the particular species of bats studied in the experiment.

In summary, please provide statistical values between groups of bats.

When supplemented with the above information, the article meets the requirements of printing in IJMS.

Author Response

We want to clarify that all bats in our study were captured in the wild and immediately euthanized on-site. Then, the dietary information for each species was derived from existing literature on these species. For this reason, this methodological approach, conducting statistical analyses between groups was not possible. Additionally, the provided details on the diet of each species are characteristics known based on the literature, and the absence of on-site diet analysis hinders direct statistical comparisons.

To address the reviewer's concern about statistical analysis, we emphasize that the most appropriate statistical assessments between groups have already been incorporated into Supplementary Tables 4, 5, and 6. These tables are specifically related to Principal Component Analysis (PCA) and offer a comprehensive and relevant exploration of group variations. We hope this clarifies the study's methodology and the rationale behind the chosen statistical analyses.

This manuscript is a resubmission of an earlier submission. The following is a list of the peer review reports and author responses from that submission.

Round 1

Reviewer 1 Report

Comments and Suggestions for Authors

This study aimed to compare oxidative metabolism in four neotropical bat species with different feeding habits and explore the connection between feeding habits and oxidative status. Significant differences were found in oxidative damage among the bat species. Antioxidant and non-antioxidant enzyme activities varied in the heart, liver, and kidney among the species. Correlation analysis between oxidative markers and antioxidants in these organs revealed diverse patterns due to varying defense mechanisms.

This paper interestingly analyzes the difference in antioxidant system according to the eating habits of bats, and the rarity of the study is acknowledged. However, some additional information is requested to aid the reader's understanding.

1.    Please provide basic information about the animal, such as the body weight of the four groups of bats and the tissue/weight ratio of the heart, liver, and kidneys, along with representative photos of each group.

2.    Is there a difference in the life expectancy of bats in the four groups due to differences in diet?

3.    Are there any reports or genetic databases on antioxidant systems or antioxidant enzymes that differentiate bats from other mammals? If there are any differences from other mammals, it will be useful information for the reader.

Author Response

This study aimed to compare oxidative metabolism in four neotropical bat species with different feeding habits and explore the connection between feeding habits and oxidative status. Significant differences were found in oxidative damage among the bat species. Antioxidant and non-antioxidant enzyme activities varied in the heart, liver, and kidney among the species. Correlation analysis between oxidative markers and antioxidants in these organs revealed diverse patterns due to varying defense mechanisms.

This paper interestingly analyzes the difference in antioxidant system according to the eating habits of bats, and the rarity of the study is acknowledged. However, some additional information is requested to aid the reader's understanding.

Reply: We thank the reviewer for taking the time to review our manuscript and especially for providing relevant suggestions. Please see below our point-to-point reply to each question/suggestion.

  1. Please provide basic information about the animal, such as the body weight of the four groups of bats and the tissue/weight ratio of the heart, liver, and kidneys, along with representative photos of each group.

Reply: We included this additional information in Table 1. Only photos were not included. In all our previous papers (including the one recently published in IJMS https://www.mdpi.com/1422-0067/24/15/12162), we never included photos of the bats.

  1. Is there a difference in the life expectancy of bats in the four groups due to differences in diet?

      Reply: Very interesting question, which we missed addressing in the manuscript. Yes, the life expectancy of bats can vary due to their diet. Different species of bats have different dietary preferences, and their diet can impact their overall health and lifespan. Bats that consume a variety of nutritionally rich diets tend to have longer lifespans than those with limited or poor-quality food sources. We added a paragraph to the manuscript addressing this issue in detail (lines 318-341).

  1. Are there any reports or genetic databases on antioxidant systems or antioxidant enzymes that differentiate bats from other mammals? If there are any differences from other mammals, it will be useful information for the reader.

      Reply: As far as we know, unfortunately, there is not.

Reviewer 2 Report

Comments and Suggestions for Authors

The present study aims to show the differences in oxidative stress between 4 species of bats, which would be linked to their feeding habits. Several points need to be improved, especially from a methodological point of view

How can we attribute the fact that this is linked to the diet and not to the species of cat? There are few individuals and therefore a significant bias which may explain the differences observed.

The figures are difficult to understand: you need to add dots for the box plots, but also show the tests and p-values, as letters are not enough. The numbers should also appear in the graphs, and the legends of all the graphs should be improved.

The structure of the results section is not logical: why end with a PCA when the PCA is a global analysis and only then do we analyze the parameters one by one, there's no logic in ending with a PCA?

Figure 4 is difficult to understand: why highlight 3 PCAs and not just one? The loadings plots should also be shown, along with a rigorous analysis of the results.

Figure 5 is poorly constructed: the left-hand side of the figure serves no purpose, and all the p-values should be shown with a legend. It's not enough just to say that it's less than 0.05.

Authors state that there is no bias related to capture and mode of analysis, these results should be shown

Additional data should be added in the main manuscript, it is still data on method and statistics

I don't understand the point of figure 6 in the manuscript, it may be an additional figure but it's irrelevant.

Comments on the Quality of English Language

Must be improved

Author Response

The present study aims to show the differences in oxidative stress between 4 species of bats, which would be linked to their feeding habits. Several points need to be improved, especially from a methodological point of view.

Reply: We thank the reviewer for taking the time to review our manuscript and especially for providing relevant suggestions. Please see below our point-to-point reply to each question/suggestion.

How can we attribute the fact that this is linked to the diet and not to the species of cat? There are few individuals and therefore a significant bias which may explain the differences observed.

Reply: Each feeding guild (insectivorous, frugivorous, etc.) is composed of different species of bats, but the similar lifestyle and dietary needs within each guild make them have similar features, like life expectancy. The life expectancy of bats with different feeding guilds can vary due to their diet. We have added an extensive paragraph explaining this matter in the manuscript (lines 318-341), which is another compelling argument that our results and conclusions are linked to diet and not to the species.

The figures are difficult to understand: you need to add dots for the box plots, but also show the tests and p-values, as letters are not enough. The numbers should also appear in the graphs, and the legends of all the graphs should be improved.

Reply: Figures 1, 2, and 3 (and their legends), are similar to our previous articles, including the recently published one in this journal on the subject (https://www.mdpi.com/1422-0067/24/15/12162), so we rather keep the standard.

The structure of the results section is not logical: why end with a PCA when the PCA is a global analysis and only then do we analyze the parameters one by one, there's no logic in ending with a PCA?

Reply: Indeed, it is better to start presenting the results section with the PCA. We have made this change in the manuscript.

Figure 4 is difficult to understand: why highlight 3 PCAs and not just one? The loadings plots should also be shown, along with a rigorous analysis of the results.

Reply: We highlight 3 PCAs and not just one because joining all these data in 1 PCA would be a mess to understand. In addition, the correlation analysis between oxidative markers and antioxidants ratified that it is better to show them separately. But we agree with the inclusion of a rigorous analysis of the results (as we already did in our previous work https://www.mdpi.com/1422-0067/24/15/12162), so we included it in the manuscript and in Table S1, Table S2, and Table S3.

Figure 5 is poorly constructed: the left-hand side of the figure serves no purpose, and all the p-values should be shown with a legend. It's not enough just to say that it's less than 0.05.

Reply: This figure is usually presented this way in most manuscripts using correlation analysis. To keep the same standard of our recent work on the subject (work https://www.mdpi.com/1422-0067/24/15/12162), we rather keep the figure as it is. However, we included supplementary tables showing all the p-values (Table S4, Table S5, and Table S6).

Authors state that there is no bias related to capture and mode of analysis, these results should be shown. Additional data should be added in the main manuscript, it is still data on method and statistics

Reply: In fact, we didn’t say there is no bias. We said that we ensured uniformity in the stressors applied, reducing the potential for significant intraspecies differences that could have arisen from these procedures (meaning, any bias, even if reduced, would be the same in all, which would not affect the results).

I don't understand the point of figure 6 in the manuscript, it may be an additional figure but it's irrelevant.

Reply: Agreed. Figure 6 was included as a supplementary figure (Fig S1).

Round 2

Reviewer 2 Report

Comments and Suggestions for Authors

The authors do not wish to respond to certain points by referring to another article published by their team in another journal. This kind of response is very surprising. As a reviewer, I tell them that I'm having trouble reading certain figures, and they ignore the point by referring to a previously published article. If, as a reviewer, I had trouble reading the figure, so will the readers.

Comments on the Quality of English Language

RAS